# Effects of temperature dependent viscosity and thermal conductivity on natural convection flow along a curved surface in the presence of exothermic catalytic chemical reaction

Uzma Ahmad[1], Muhammad Ashraf[1], A. Al-Zubaidi[3], Aamir Ali[2]*, Salman Saleem[3]

1 Department of Mathematics, Faculty of Science, University of Sargodha, Sargodha, Pakistan,
2 Department of Mathematics, COMSATS University Islamabad, Attock, Pakistan, 3 Department of Mathematics, College of Science, King Khalid University, Abha, Saudi Arabia

* aamir_ali@cuiatk.edu.pk

**Data Availability Statement:** All relevant data are within the paper files.

## Abstract

The natural convection boundary layer flow of a viscous incompressible fluid with temperature dependent viscosity and thermal conductivity in the presence of exothermic catalytic chemical reaction along a curved surface has been investigated. The governing non dimensional form of equations is solved numerically by using finite difference scheme. The numerical results of velocity profile, temperature distribution and mass concentration as well as for skin friction, heat transfer rate and mass transfer rate are presented graphically and in tabular form for various values of dimensionless parameters those are generated in flow model during dimensionalization. From the obtained results, it is concluded that the exothermic catalytic chemical reactions is associated with temperature dependent viscosity and thermal conductivity. Further, it is concluded that the body shape parameter also plays an important quantitative role for change in velocity profile, temperature field and mass concentration behavior in the presence of exothermic catalytic chemical reaction.

## Introduction

The idea of natural convection heat transfer along the curved surface in the presence of exothermic catalytic chemical reaction represents an idealization of many significant issues in engineering practice. Because of the complexities involved in formulation and solving these problems, it is common to neglect the effect of variable viscosity and thermal conductivity, and consider only relatively low temperature problems. In this study, we consider complex geometry and very complex mechanism in terms of natural convection heat transfer. Carey and Mollendorf [1] gave the similarity analysis for the liquids that have larger viscosity variations as compared to the variations in other fluid properties. The results showed that viscosity varied linearly with temperature for extensive choices of Prandtl number. The effects of natural convective flow over a body of arbitrary geometric shape were numerically demonstrated by

**Funding:** The authors extend their appreciation to the Deanship of Scientific Research at King Khalid University for funding this work through research groups program under Grant No. RGP.1/46/42.

**Competing interests:** The authors have declared that no competing interests exist.

Pop and Takhar [2]. It was observed that the local Nusselt number dropped down with the increasing body shape parameter $n$ and enhanced with increasing Pr, whereas the reverse behavior was shown by velocity and temperature profiles. Merkin and Chaudhary [3–5] elaborated the effects of catalytic exothermic chemical reactions on a vertical surface and near a stagnation point. Hossain *et al.* [6] presented the effects of temperature dependent viscosity graphically for free convective flow about a truncated cone. In [7, 8], a comprehensive analytical approach has been carried out by Hossain *et al.* on free convection flow about an isothermal vertical wavy cone and a truncated cone under the influence of variable viscosity and thermal conductivity. In [8], the authors implemented the perturbation technique to find the solutions near and far away from the point of truncation. A brief note on the free convection from curved surfaces was presented by Magyari *et al.* [9]. They summarized that no free convection can occur below the infinite surface. Taking under consideration the effect of variable viscosity and thermal conductivity, Hossain *et al.* [10, 11] investigated the natural convection boundary layer flow from a vertical wavy surface having uniform temperature and the forced-free convection along a vertical flat plate. A numerical and theoretical study of natural convection flow have been done by Molla *et al.* [12, 13] along an isothermal sphere under the effect of temperature dependent thermal conductivity and from an isothermal horizontal circular cylinder by considering the viscosity inversely proportional to the temperature by using both implicit finite difference scheme and the Keller box scheme. Later, Rahman *et al.* [14] and Molla *et al.* [12, 13] approximated the effect of temperature dependent thermal conductivity on a vertical flat plate.

Uddin and Kumar [15] critically discussed the behavior of electrically conducting natural convection fluid flow near the lower stagnation point of an isothermal porous cylinder by assuming the viscosity and thermal conductivity to be temperature reliant. They concluded that both the temperature and velocity increased by raising the thermal conductivity parameter but both have shown an opposite behavior for variable viscosity parameter. Mahanti and Gaur [16] reported the same result as in [15] for thermal conductivity parameter but an adverse behavior was observed for variable viscosity parameter. An increase in viscosity parameter decreased the fluid velocity near the plate and showed negligible effect on temperature. Modather *et al.* [17] presented their findings for an unsteady two-dimensional laminar flow of a micropolar fluid along a horizontal stretching sheet in the presence of temperature dependent properties of the fluid. Thohura *et al.* [18] studied the free convective flow with temperature dependent thermal conductivity and heat flux, and found out that with the increasing values of thermal conductivity variation parameter, the velocity and temperature of the fluid have increased considerably, resulting in the increase of thermal boundary layer thickness. Umavathi *et al.* [19] briefly described the combined effects of viscosity and variable thermal conductivity for steady natural convection flow both analytically and numerically by using perturbation and fourth order Runge-Kutta method. The effects of temperature dependent viscosity and thermal conductivity for free- forced convection flow has been keenly studied by Ashraf *et al.* [20] along a magnetized vertical surface. Makanda [21] discussed the free convection flow around a spinning sphere by considering that viscosity, thermal conductivity and viscous dissipation are temperature dependent. He solved the partial differential equations by a newly developed spectral method called bi-variate local linearization method. Keimanesh and Aghanajafi [22] examined the effects of radiation heat flux, magnetic field and the porous sheet for a micropolar fluid flow over a stretching sheet. They considered the viscosity and thermal conductivity as temperature- dependent. The results declare a decrease in fluid velocity due to magnetic field and porousness, but the radiation heat flux enhances the boundary layer thickness. Anantha *et al.* [23–30] studied the mechanism of heat and mass transfer for different characteristics of fluid around different geometries. Lately, Ahraf *et al.* [31, 32]

analyzed the natural convection heat and mass transfer flow along a curved surface in the presence of exothermic catalytic chemical reaction.

In current paper we are investigated the natural convection flow over a two dimensional body of arbitrary geometric configuration in the presence of exothermic catalytic chemical reaction. The momentum, energy, and mass concentration equations are a general form suitable for laminar natural convection flows along curved surface in the inclusion of exothermic catalytic reaction. For body of arbitrary shape, the special case in which $P(x)+Q(x) = n$, thus the body shape parameter (index parameter) $n$ has chosen (0, 1/2] which is main novelty of this work. The proposed model is used as the guidelines for the selection of a suitable coupling to achieve the desired applications. The curved shaped geometry is used to design many problems of civil engineering as pressure barrier. The overall objectives of this research are to develop the mathematical model to study and compare the different modes of coupling the exothermic and catalytic chemical reaction via momentum, energy and mass concentration equation. Model predictions are used to assess the effects of different parameters on conversion of exothermic catalytic chemical reaction at heated curve generated by tangential component of acceleration due to gravity. The proposed model is used as the guidelines for the selection of a suitable coupling to achieve the desired applications. A numerical technique Finite Difference Method in conjunction with primitive variable formulation is used to investigate the coupling of exothermic reaction with catalyst particles. Furthermore, parametric effects of heated wall and mass concentration along the curved surface are studied and highlighted graphically and as well as in tabular form.

## The problem and governing equations

A typical flow configuration is illustrated in Fig 1, where $x$ is the coordinate along the curved surface and $y$ normal to it. A two dimensional, steady, laminar free convection flow of a

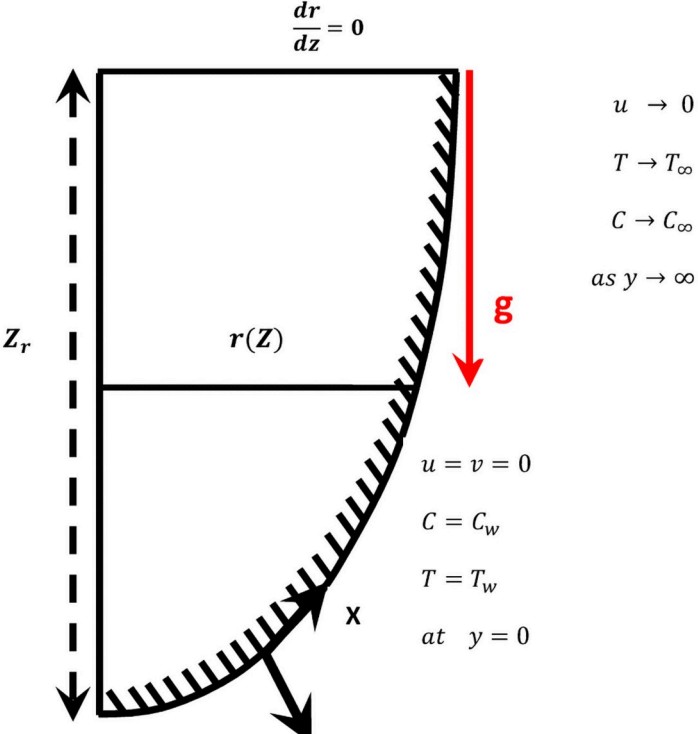

**Fig 1. Flow geometry and the coordinate system.**

viscous, incompressible fluid along a curved surface under the influence of exothermic cata-
lytic chemical reaction is considered. The temperature of the surface, $T_w$, is higher than the
ambient fluid temperature $T_\infty$.

Moreover, μ and $k$ are assumed to be temperature dependent and can be defined as follows:

$$\mu = \mu_\infty(1 + \gamma^*(T - T_\infty)), \tag{1a}$$

$$k = k_\infty(1 + \xi^*(T - T_\infty)). \tag{1b}$$

Where $\mu_\infty$ is the viscosity and $k_\infty$ is the thermal conductivity of the ambient fluid away from
the surface $\gamma^*$ and $\xi^*$ are the constants.

Incorporating the usual Boussinesq approximation within the boundary layer, the flow gov-
erned by the following dimensionless form of the continuity, momentum, energy and the mass
concentration by following [31] are:

$$\frac{\partial \bar{u}}{\partial \bar{x}} + \frac{\partial \bar{v}}{\partial \bar{y}} = 0, \tag{2}$$

$$\bar{u}\frac{\partial \bar{u}}{\partial \bar{x}} + \bar{v}\frac{\partial \bar{u}}{\partial \bar{y}} + \frac{\bar{u}^2}{2g_x}\frac{dg_x}{d\bar{x}} = \gamma_\mu\left(\frac{\partial \theta}{\partial \bar{y}}\right)\left(\frac{\partial \bar{u}}{\partial \bar{y}}\right) + \left(1 + \gamma_\mu\theta\right)\left(\frac{\partial^2 \bar{u}}{\partial \bar{y}^2}\right) + \theta + \phi, \tag{3}$$

$$\bar{u}\frac{\partial \theta}{\partial \bar{x}} + \bar{v}\frac{\partial \theta}{\partial \bar{y}} = \frac{1}{pr}\left\{\xi\left(\frac{\partial \theta}{\partial \bar{y}}\right)^2 + (1 + \xi\theta)\frac{\partial^2\theta}{\partial \bar{y}^2}\right\} + \beta\lambda^2(1 + n\gamma\theta)exp\left(\frac{-E}{1 + \gamma\theta}\right)\theta, \tag{4}$$

$$\bar{u}\frac{\partial \phi}{\partial \bar{x}} + \bar{v}\frac{\partial \phi}{\partial \bar{y}} = \frac{1}{sc}\frac{\partial^2 \phi}{\partial \bar{y}^2} + \lambda^2(1 + n\gamma\theta)exp\left(\frac{-E}{1 + \gamma\theta}\right)\phi. \tag{5}$$

The appropriate dimensionless boundary conditions at the surface and far from the surface
are

$$\bar{u} = 0, \quad \bar{v} = 0, \quad \theta = 1, \quad \phi = 1 \quad at \quad \bar{y} = 0,$$
$$\bar{u} \to 0, \quad \theta \to 0, \quad \phi \to 0 \quad as \quad \bar{y} \to \infty. \tag{6}$$

Here $\bar{u}$ and $v$ denote the dimensionless fluid velocity components in $x$- and $y$- directions
respectively, $\theta$ and $\phi$ are the dimensionless temperature and mass concentration, Pr and Sc are
the Prandtl and Schmidt numbers, defined as $Pr = v/\alpha$, $Sc = v/D_B$, $\gamma_\mu = \gamma^*\Delta T$ is the viscosity var-
iation parameter and $\xi = \xi^*\Delta T$ is the thermal conduction variation parameter. $\beta = 1$ is the exo-
thermic parameter, $\lambda^2 = k_r^2\delta^2/v$ the dimensionless chemical reaction rate constant, $\gamma = (T_w -
T_\infty)/T_\infty$ is the temperature relative parameter, $E = E_a/k_g(T - T_\infty)$ is the dimensionless activa-
tion energy. Moreover $g_x$ is the tangential component of acceleration due to gravity and its
presence in momentum equation defines the curved surface. It is defined as defined as

$$g_x = \left(1 - \left(\frac{dr}{dx}\right)^2\right)^{1/2}$$

Since we know that

$$\frac{d}{dx}(lnx) = \frac{1}{x}$$

and

$$d(lnx) = \frac{1}{x} dx$$

which implies that

$$\frac{1}{d(lnx)} = \frac{x}{dx}. \tag{7}$$

Also consider

$$\frac{d}{dx}(ln \ g_x) = \frac{1}{g_x}\frac{dg_x}{dx}$$

which implies

$$d(ln \ g_x) = \frac{1}{g_x}dg_x. \tag{8}$$

Now combining the above two Eqs (7) and (8), we have

$$\frac{d \ ln \ g_x}{d \ ln \ x} = \frac{x}{g_x}\frac{dg_x}{dx}.$$

In dimensionless form, it is

$$\frac{d \ lng_x}{d \ ln\bar{x}l} = \frac{\bar{x}}{g_x}\frac{dg_x}{d\bar{x}}$$

and thus we have

$$\frac{1}{\bar{x}}\frac{d \ lng_x}{d \ ln\bar{x}l} = \frac{1}{g_x}\frac{dg_x}{d\bar{x}}. \tag{9}$$

Also, we consider

$$\frac{d}{dx}(lnT_w) = \frac{1}{T_w}\frac{dT_w}{dx} = 0$$

which implies that

$$d \ lnT_w = 0$$

and thus, we can write

$$\frac{d \ ln \ T_w}{d \ ln \ \bar{x}l} = 0 \tag{10}$$

Now from Eqs (9) and (10), we have

$$1\bar{x}\left(\frac{d \ lnT_w}{d \ ln\bar{x} \ l} + \frac{d \ lng_x}{d \ ln\bar{x} \ l}\right) = \frac{1}{g_x}\frac{dg_x}{d\bar{x}}$$

which can also be written as

$$\frac{1}{\bar{x}}(P(\bar{x}) + Q(\bar{x})) = \frac{1}{g_x}\frac{dg_x}{d\bar{x}} \tag{11}$$

Here, $P(\bar{x})$ the wall temperature function and $Q(\bar{x})$ the body shape function, are defined as

$$P(\bar{x}) = \frac{d \; lnT_w}{d \; ln\bar{x}\imath}, \quad Q(\bar{x}) = \frac{d \; lng_x}{d \; ln\bar{x}\imath}.$$

For the special case (see [2]) in which both $P(x)$ and $Q(x)$ are constants namely, $m_1$ and $n_1$, which satisfy the relation

$$m_1 + n_1 = n \tag{12}$$

Under the conditions given in (12), the conservation equations along with boundary conditions (by dropping bars) take the following form

$$\frac{\partial u}{\partial x} + \frac{\partial v}{\partial y} = 0, \tag{13}$$

$$u\frac{\partial u}{\partial x} + v\frac{\partial u}{\partial y} + \frac{u^2}{2x}n = \gamma_\mu \left(\frac{\partial \theta}{\partial y}\right)\left(\frac{\partial u}{\partial y}\right) + (1 + \gamma_\mu\theta)\left(\frac{\partial^2 u}{\partial y^2}\right) + \theta + \phi, \tag{14}$$

$$u\frac{\partial \theta}{\partial x} + v\frac{\partial \theta}{\partial y} = \frac{1}{Pr}\left\{\xi\left(\frac{\partial \theta}{\partial y}\right)^2 + (1 + \xi\theta)\frac{\partial^2 \theta}{\partial y^2}\right\} + \beta\lambda^2(1 + n\gamma\theta)exp\left(\frac{-E}{1 + \gamma\theta}\right)\theta, \tag{15}$$

$$u\frac{\partial \phi}{\partial x} + v\frac{\partial \phi}{\partial y} = \frac{1}{sc}\frac{\partial^2 \phi}{\partial y^2} + \lambda^2(1 + n\gamma\theta)exp\left(\frac{-E}{1 + \gamma\theta}\right)\phi. \tag{16}$$

with dimensionless boundary conditions

$$\begin{aligned} u = 0, \quad v = 0, \quad \theta = 1, \quad \varphi = 1 \quad at \quad y = 0 \\ u \to 0, \quad \theta \to 0, \quad \phi \to 0 \quad as \quad y \to \infty. \end{aligned} \tag{17}$$

## Numerical methods

To solve the Eqs (13)–(16) subject to the boundary conditions (17), we transform these equations into primitive form by introducing the following dimensionless primitive variable transformations:

$$\begin{aligned} u = x^{1/2}U(X, Y), \quad v = x^{-1/2}V(X, Y), \quad x = X, \quad y = x^{1/4}Y, \\ \theta = \Theta(X.Y), \quad\quad C = \Phi(X, Y). \end{aligned} \tag{18}$$

After using the aforementioned transformations, we have the following form of system of equations:

$$\frac{U}{2} + X\frac{\partial U}{\partial x} - \frac{Y}{4}\frac{\partial U}{\partial Y} + \frac{\partial V}{\partial Y} = 0, \tag{19}$$

$$\left[\frac{1}{2} + \frac{n}{2}\right]U^2 + XU\frac{\partial U}{\partial X} + \left(V - \frac{YU}{4}\right)\frac{\partial U}{\partial Y} = \gamma_\mu\frac{\partial \Theta}{\partial Y}\frac{\partial U}{\partial Y} + \left(1 + \gamma_\mu\Theta\right)\frac{\partial^2 U}{\partial Y^2} + \Theta + \Phi, \tag{20}$$

$$XU\frac{\partial \Theta}{\partial X} + \left(V - \frac{YU}{4}\right)\frac{\partial \Theta}{\partial Y} = \frac{1}{Pr}\left[\xi\left(\frac{\partial \Theta}{\partial Y}\right)^2 + (1 + \xi\Theta)\frac{\partial^2 \Theta}{\partial Y^2}\right] + \beta\lambda^2 X^{1/2}(1 + n\gamma\Theta)exp\left(\frac{-E}{1 + \gamma\theta}\right)\Theta, \tag{21}$$

$$XU\frac{\partial \Phi}{\partial X} + \left(V - \frac{YU}{4}\right)\frac{\partial \Phi}{\partial Y} = \frac{1}{Sc}\frac{\partial^2 \Phi}{\partial Y^2} + \lambda^2 X^{1/2}(1 + n\gamma\Theta)exp\left(\frac{-E}{1 + \gamma\theta}\right)\Phi. \tag{22}$$

The boundary conditions will take the form

$$\begin{aligned} U = 0, \quad V = 0, \quad \theta = 1, \quad \Phi = 1, \quad at \quad Y = 0; \\ U \to 0, \quad \Theta \to 0, \quad \Phi \to 0 \quad as \quad Y \to \infty. \end{aligned} \tag{23}$$

**Discretization.** We now discretized the above transformed boundary layer equations by using Finite Difference Method (FDM). We applied backward difference along x-axis and central difference along y-axis.

$$\frac{\partial U}{\partial X} = \frac{U_{i,j} - U_{i,j-1}}{\Delta X}, \quad \frac{\partial U}{\partial Y} = \frac{U_{i+1,j} - U_{i-1,j}}{2\Delta Y}, \quad \frac{\partial^2 U}{\partial Y^2} = \frac{U_{i+1,j} - 2U_{i,j} + U_{i-1,j}}{\Delta Y^2} \tag{24}$$

Thus by employing (24), the discretized form of momentum, energy and mass concentration Eqs (19)–(22) along with boundary conditions (23) is given below

$$A_1 U_{i-1,j} + B_1 U_{i,j} + C_1 U_{i+1,j} = D_1, \tag{25}$$

where

$$A_1 = \frac{\Delta Y}{2}\left(V_{i,j} - \frac{U_{i,j}Y_j}{4}\right) - \frac{1}{4}\gamma_\mu\left(\Theta_{i+1,j} - \Theta_{i-1,j}\right) + \left(1 + \gamma_\mu\Theta_{i,j}\right),$$

$$B_1 = -\left[\frac{1}{2} + \frac{n}{2} + \frac{X_i}{\Delta X}\right]\Delta Y^2 U_{i,j} - 2\left(1 + \gamma_\mu\Theta_{i,j}\right),$$

$$C_1 = -\left[\frac{\Delta Y}{2}\left(V_{i,j} - \frac{U_{i,j}Y_j}{4}\right) - \frac{1}{4}\gamma_\mu\left(\Theta_{i+1,j}\right)\right] + \left(1 + \gamma_\mu\Theta_{i,j}\right),$$

$$D_1 = -\frac{X_i}{\Delta X}\Delta Y^2 U_{i,j}U_{i,j-1} - \Delta Y^2\Theta_{i,j} - \Delta Y^2\Phi_{i,j}.$$

$$A_2\theta_{i-1,j} + B_2\theta_{i,j} + C_2\theta_{i+1,j} = D_2, \tag{26}$$

where

$$A_2 = \frac{\Delta Y}{2}\left(V_{i,j} - \frac{U_{i,j}Y_j}{4}\right) - \frac{1}{Pr}\frac{\xi}{4}\left(\Theta_{i+1,j} - \Theta_{i-1,j}\right) + \frac{1}{Pr}\left(1 + \xi\Theta_{i,j}\right),$$

$$B_2 = -\frac{X_i}{\Delta x}\Delta Y^2 U_{i,j} - \frac{2}{pr}\left(1 + \xi\Theta_{i,j}\right) + \beta X_i^{1/2}\lambda^2\Delta y^2\left(1 + n\gamma\Theta_{i,j}\right)e^{\frac{-E}{1+\gamma\Theta_{i,j}}},$$

$$C_2 = -\left[\frac{\Delta Y}{2}\left(V_{i,j} - \frac{U_{i,j}Y_j}{4}\right) - \frac{1}{pr}\frac{\xi}{4}\left(\Theta_{i+1,j} - \Theta_{i-1,j}\right)\right] + \frac{1}{pr}\left(1 + \xi\Theta_{i,j}\right),$$

$$D_2 = -\frac{X_i}{\Delta X}\Delta Y^2 U_{i,j}\Theta_{i,j-1}.$$

$$A_3\Phi_{i-1,j} + B_3\Phi_{i,j} + C_3\Phi_{i+1,j} = D_3, \tag{27}$$

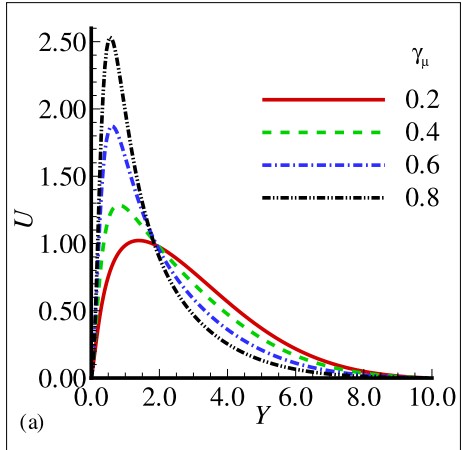
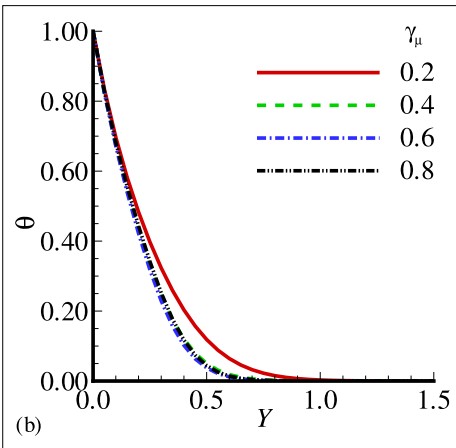

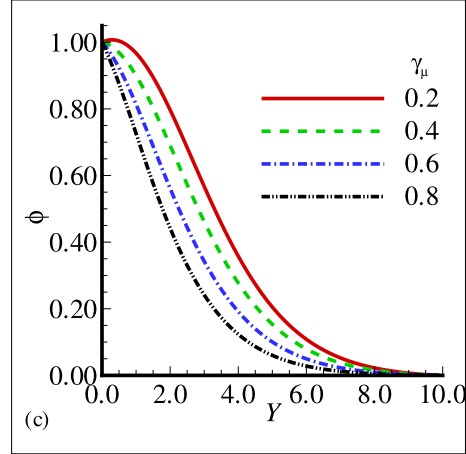

**Fig 2.** Profiles for (a) $U$ (b) $\theta$ and (c) $\Phi$ for different viscosity variation parameter $\gamma_\mu$ with $n = 0.3$, $Pr = 7.0$, $Sc = 0.8$, $E = 0.2$, $\beta = 0.8$, $\gamma = 0.8$, $\lambda = 0.5$ and $\xi = 0.1$.

where

$$A_3 = \frac{\Delta Y}{2}\left(V_{i,j} - \frac{U_{i,j}Y_j}{5}\right) + \frac{1}{Sc},$$

$$B_3 = -\frac{x_i}{\Delta x}\Delta Y^2 U_{i,j} - \frac{2}{sc} + X_i^{1/2}\lambda^2\Delta y^2\left(1 + n\gamma\Theta_{i,j}\right)e^{\frac{-E}{1+\gamma\theta_{ij}}},$$

$$C_3 = -\frac{\Delta Y}{2}\left(V_{i,j} - \frac{U_{i,j}Y_j}{5}\right) + \frac{1}{Sc},$$

$$D_3 = -\frac{X_i}{\Delta x}\Delta Y^2 U_{i,j}\Phi_{i,j-1}.$$

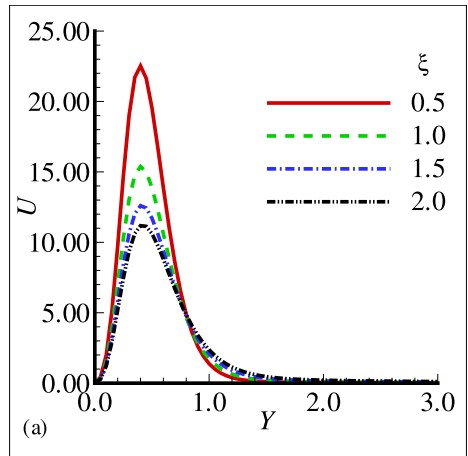

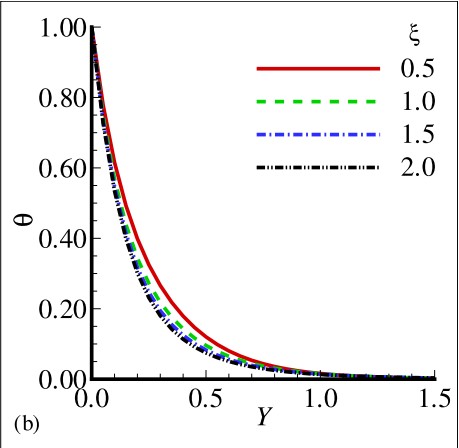

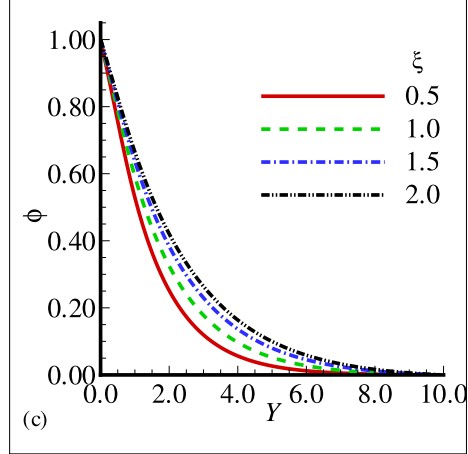

**Fig 3.** Profiles for (a) $U$ (b) $\theta$ and (c) $\Phi$ for different thermal conductivity variation parameter $\xi$ with $n = 0.5$, $Pr = 0.5$, $Sc = 0.1$, $E = 0.2$, $\beta = 0.4$, $\gamma = 0.2$, $\lambda = 0.5$ and $\gamma_\mu = 0.8$.

The discretized boundary conditions are

$$U_{i,j} = 0, \quad V_{i,j} = 0, \quad \Theta_{i,j} = 1, \quad \Phi_{i,j} = 1 \quad at \quad Y_j = 0$$
$$U_{i,j} \to 0, \quad \theta_{i,j} \to 0, \quad \Phi_{i,j} \to 0 \quad as \quad Y_j \to \infty$$

(28)

The Gaussian elimination method was implemented to solve tri-diagonal set of algebraic Eqs (25)–(27) along with boundary conditions (28). To reduce the error from a predefined error an iterative solution is performed for all the solution variables U, V, range and size. The effects of each variable is carried out. The directions along and normal to the surface are represented by the subscripts $i, j$.

The velocity field, skin friction, rate of heat and mass transfer are calculated as follows

$$V_{i+1,j} = V_{i-1,j} - \frac{6}{5}\Delta Y U_{i,j} - 2\frac{\Delta Y}{\Delta X}X_i\left(U_{i,j} - U_{i,j-1}\right) + \frac{Y_j}{5}\left(U_{i+1,j} - U_{i-1,j}\right),$$
$$\tau_w = \left(\frac{\partial U}{\partial Y}\right)_{y=0,} \quad \theta_w = \left(\frac{\partial \theta}{\partial Y}\right)_{y=0}, \quad \phi_w = \left(\frac{\partial \phi}{\partial Y}\right)_{y=0}$$

(29)

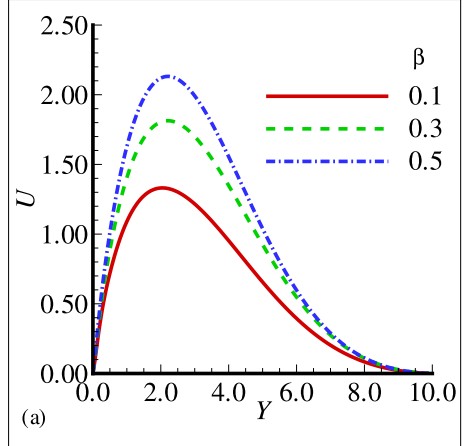

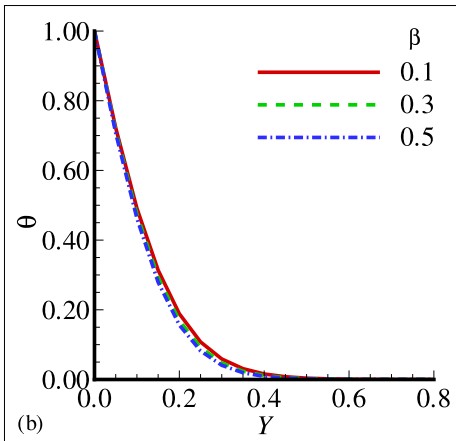

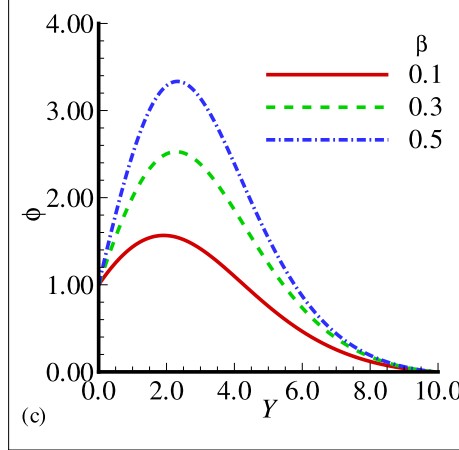

**Fig 4.** Profiles for (a) $U$ (b) $\theta$ and (c) $\Phi$ for different exothermic parameter $\beta$ with $n = 0.3$, $Pr = 0.5$, $Sc = 0.4$, $E = 0.1$, $\gamma = 0.1$, $\lambda = 0.8$, $\xi = 0.4$ and $\gamma_\mu = 0.2$.

## Numerical results and discussion

In this section, the effects of different physical parameters are obtained in the form of velocity, temperature and mass concentration and are presented graphically and through tables. The effects of viscosity variation parameter $\gamma_\mu$ on temperature, velocity and mass concentration are presented in Fig 2. It can be observed that the velocity has increased prominently near the surface as viscosity variation parameter has increased, whereas the temperature and mass concentration have decreased with the increasing $\gamma_\mu$. From the above situation, it is clear that the fluid has accelerated with the increasing values of $\gamma_\mu$. Fig 3(a) and 3(b) shows that the velocity and temperature have decreased with increasing values of thermal conductivity variation parameter $\xi$ but from Fig 3(c), we see that mass concentration has exhibited a slight increase with the increase in $\xi$. All the three figures have presented an asymptotic behavior far away from the surface as per our boundary conditions. Fig 4 displays the influence of exothermic parameter $\beta$ on velocity, temperature and mass concentration. It is observed that the temperature has not merely affected by exothermic parameter $\beta$ but the velocity and mass concentration have represented a significant increase with the increasing values of $\beta$ and showed an asymptotic behavior far away from the surface. Considering Fig 5, it is observed that increase in Prandtl

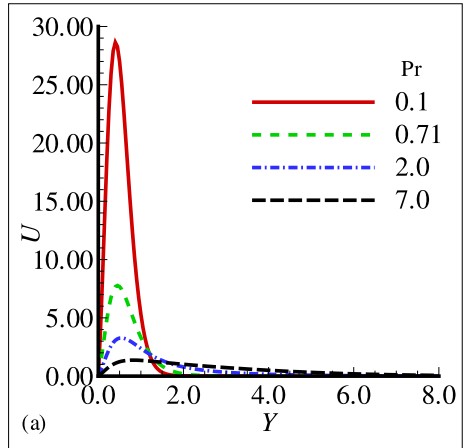
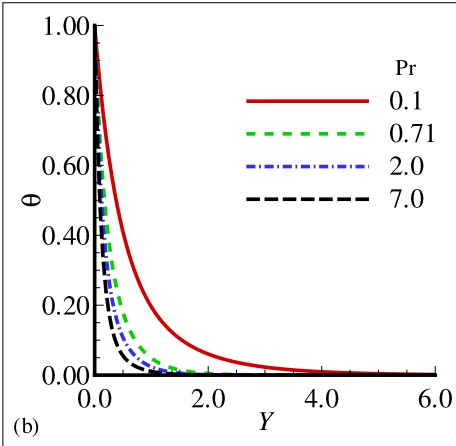

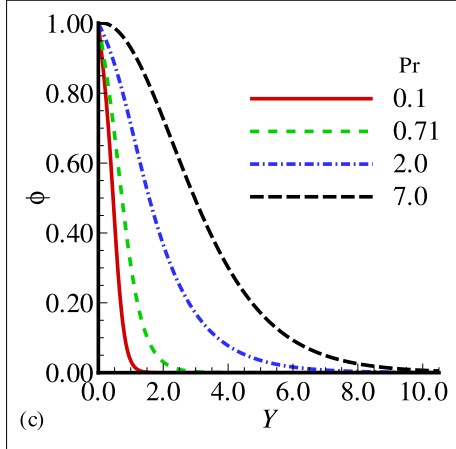

**Fig 5.** Profiles for (a) $U$ (b) $\theta$ and (c) $\Phi$ for different Prandtl number Pr with $n$ = 0.3, $Sc$ = 0.8, $E$ = 0.1, $\beta$ = 0.2, $\gamma$ = 0.5, $\lambda$ = 0.5, $\xi$ = 0.2 and $\gamma_\mu$ = 0.2.

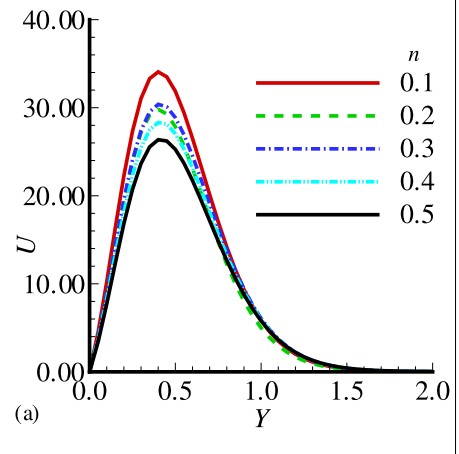
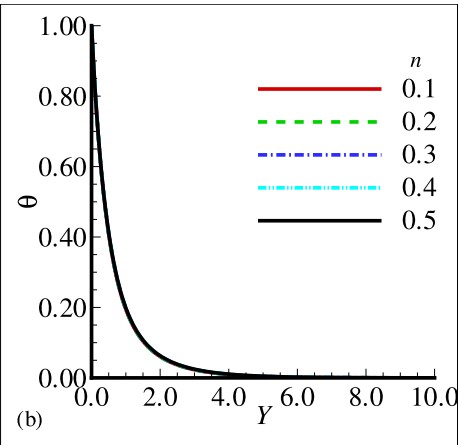

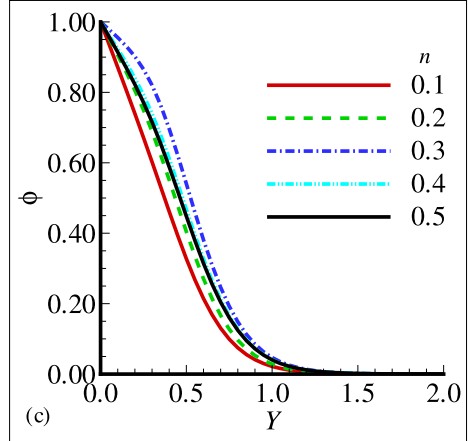

**Fig 6.** Profiles for (a) $U$ (b) $\theta$ and (c) $\Phi$ for different body shape parameter $n$ with $Pr = 7.0$, $Sc = 0.8$, $E = 0.1$, $\beta = 1.0$, $\gamma = 0.5$, $\lambda = 0.5$, $\xi = 0.2$ and $\gamma_\mu = 0.2$.

number Pr has significantly decreased the velocity and temperature but a totally reverse profile has noted for mass concentration. The reason is that with the increase of Pr, the fluid viscosity grows, which in result de-accelerate the fluid motion. Also, since Pr is inversely proportional to the thermal conductivity, increase in Pr decreases the temperature. Fig 6a–6c exhibit a gradual decrease in velocity and a gradual increase in mass concentration as we increase the values of body shape parameter $n$ but do not reveal any change in the behavior of the temperature.

The influence of viscosity variation parameter $\gamma_\mu$ on $\frac{\partial U}{\partial Y}$, $\frac{\partial \theta}{\partial Y}$ and $\frac{\partial \phi}{\partial Y}$ are given in Table 1. We observe from this table that, at the surface, the skin friction and the mass transfer are increasing with increasing values of $\gamma_\mu$, whereas the heat transfer has revealed an opposite behavior. Along the surface, the skin friction behaves differently for different values of $\gamma_\mu$, it is increasing for $\gamma_\mu = 0.7$, and decreasing for $\gamma_\mu = 0.8$. The heat transfer rate keeps on increasing slowly along the surface, but the mass transfer is increasing halfway and then start decreasing for both values of $\gamma_\mu$ Table 2 presents the effect of two different values of thermal conductivity variation parameter $\xi$ (i.e. 0.5 and 0.9) by keeping other parameters constant. From this table, it is clearly seen that the heat and mass rate transfer are increasing at the surface with the increasing values of $\xi$, whereas the skin friction is decreasing. Moreover, along the surface and away from the leading edge, the skin friction and mass flux lead to a decrease for $\xi = 0.5$ but represents a

**Table 1. The results of $\frac{\partial U}{\partial Y}$, $\frac{\partial \theta}{\partial Y}$ and $\frac{\partial \phi}{\partial Y}$ for different $\gamma_\mu$ by keeping $n = 0.5$, $Pr = 5.0$, Sc = 0.8, $E = 0.1$, $\beta = 0.8$, $\gamma = 0.8$, $\lambda = 0.2$ and $\xi = 0.8$.**

| | $\left(\frac{\partial U}{\partial Y}\right)_{y=0}$ | | $\left(\frac{\partial \Theta}{\partial Y}\right)_{y=0}$ | | $\left(\frac{\partial \Phi}{\partial Y}\right)_{y=0}$ | |
|---|---|---|---|---|---|---|
| X | $\gamma_\mu = 0.7$ | $\gamma_\mu = 0.8$ | $\gamma_\mu = 0.7$ | $\gamma_\mu = 0.8$ | $\gamma_\mu = 0.7$ | $\gamma_\mu = 0.8$ |
| 0.0 | -0.03263 | 0.11762 | 8.31615 | 8.31614 | 0.06094 | 0.07477 |
| 0.1 | -0.03263 | 0.11762 | 8.31615 | 8.31614 | 0.06094 | 0.07477 |
| 1.0 | 0.52240 | 0.08302 | 8.31579 | 8.31581 | 0.45930 | 0.48160 |
| 2.0 | 0.59996 | 0.07747 | 8.31584 | 8.31585 | 0.46806 | 0.49869 |
| 3.0 | 0.63072 | 0.07522 | 8.31585 | 8.31587 | 0.46889 | 0.50242 |
| 4.0 | 0.64824 | 0.07393 | 8.31586 | 8.31588 | 0.46808 | 0.50302 |
| 5.0 | 0.65988 | 0.07306 | 8.31587 | 8.31588 | 0.46687 | 0.50263 |
| 6.0 | 0.66832 | 0.07243 | 8.31587 | 8.31588 | 0.46560 | 0.50188 |
| 7.0 | 0.67480 | 0.07195 | 8.31587 | 8.31589 | 0.46439 | 0.50102 |
| 8.0 | 0.67998 | 0.07156 | 8.31588 | 8.31589 | 0.46325 | 0.50013 |
| 9.0 | 0.68424 | 0.07124 | 8.31588 | 8.31589 | 0.46220 | 0.49926 |
| 10.0 | 0.68782 | 0.07097 | 8.31588 | 8.31589 | 0.46122 | 0.49841 |

**Table 2. The results of $\frac{\partial U}{\partial Y}$, $\frac{\partial \theta}{\partial Y}$ and $\frac{\partial \phi}{\partial Y}$ for different $\xi$ by keeping $n = 0.5$, $Pr = 7.0$, Sc = 0.8, $E = 0.3$, $\beta$, $\gamma = 0.2$, $\lambda = 0.5$ and $\gamma_\mu = 0.6$.**

| | $\left(\frac{\partial U}{\partial Y}\right)_{y=0}$ | | $\left(\frac{\partial \Theta}{\partial Y}\right)_{y=0}$ | | $\left(\frac{\partial \Phi}{\partial Y}\right)_{y=0}$ | |
|---|---|---|---|---|---|---|
| X | $\xi = 0.5$ | $\xi = 0.9$ | $\xi = 0.5$ | $\xi = 0.9$ | $\xi = 0.5$ | $\xi = 0.9$ |
| 0.0 | 0.10665 | 0.06929 | 8.08107 | 8.36242 | 0.03465 | 0.19159 |
| 0.1 | 0.10665 | 0.07232 | 8.08107 | 8.36239 | 0.03465 | 0.18907 |
| 1.0 | 0.04462 | 0.07446 | 8.08038 | 8.36238 | 0.29731 | 0.19363 |
| 2.0 | 0.03073 | 0.07476 | 8.08045 | 8.36238 | 0.28549 | 0.19514 |
| 3.0 | 0.02464 | 0.07489 | 8.08048 | 8.36238 | 0.28014 | 0.19592 |
| 4.0 | 0.02102 | 0.07497 | 8.08050 | 8.36238 | 0.27677 | 0.19642 |
| 5.0 | 0.01854 | 0.07502 | 8.08051 | 8.36238 | 0.27433 | 0.19678 |
| 6.0 | 0.01670 | 0.07506 | 8.08051 | 8.36238 | 0.27246 | 0.19706 |
| 7.0 | 0.01527 | 0.07509 | 8.08052 | 8.36238 | 0.27094 | 0.19728 |
| 8.0 | 0.01411 | 0.07512 | 8.08052 | 8.36238 | 0.26968 | 0.19747 |
| 9.0 | 0.01314 | 0.07514 | 8.08053 | 8.36238 | 0.26860 | 0.19762 |
| 10.0 | 0.01233 | 0.07515 | 8.08053 | 8.36238 | 0.26767 | 0.19776 |

gradual increase for $\xi = 0.9$. On the other hand, the heat transfer is increasing along the surface for $\xi = 0.5$, but remain constant for $\xi = 0.9$.

## Conclusion

The results indicate that the velocity boosts up with an increase in viscosity variation parameter $\gamma_\mu$, whereas a decreasing behavior is observed in the temperature and mass concentration. The maximum velocity decreases as thermal conductivity variation parameter $\xi$ increases, whereas the temperature represents a gradual decrease and the mass concentration, a gradual increase with increasing $\xi$. Further, the exothermic parameter $\beta$ influences the temperature negligibly. The velocity and mass transfer rate show a considerable increase for increasing $\beta$. The velocity and temperature decrease, whereas the mass concentration shows a gradual increase as the Prandtl number $Pr$ increases. For enhancing values of body shape parameter $n$, the mass concentration enhances but the velocity diminishes. No change is observed in case of

temperature. The increase in viscosity variation parameter $\gamma_\mu$, increases the heat transfer along the surface, fluctuates the mass transfer midway and show opposite trends in case of skin friction for both values of $\gamma_\mu$. For thermal conductivity variation parameter $\xi$, the heat transfer of the fluid molecules is not affected for $\xi = 0.9$, but it increases along the surface for $\xi = 0.5$. The skin friction is increasing along the surface for both $\xi$, but mass flux behaves differently for both values.

## Supporting information

**S1 Nomenclature.**
(DOCX)

## Author Contributions

**Formal analysis:** Uzma Ahmad.

**Funding acquisition:** A. Al-Zubaidi.

**Resources:** Salman Saleem.

**Supervision:** Muhammad Ashraf.

**Visualization:** Salman Saleem.

**Writing – review & editing:** A. Al-Zubaidi, Aamir Ali, Salman Saleem.

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
