## [Decision Letter · Decision Letter 0]

10 Mar 2021

PONE-D-21-04681

Effects of Temperature Dependent Viscosity and Thermal Conductivity on Natural Convection Flow along a Curved Surface in the Presence of Exothermic Catalytic Chemical Reaction

PLOS ONE

Dear Dr. Ali,

Thank you for submitting your manuscript to PLOS ONE. After careful consideration, we feel that it has merit but does not fully meet PLOS ONE’s publication criteria as it currently stands. Therefore, we invite you to submit a revised version of the manuscript that addresses the points raised during the review process.

We look forward to receiving your revised manuscript.

Kind regards,

Naramgari Sandeep, Ph.D

Academic Editor

PLOS ONE

Journal Requirements:

2) PLOS requires an ORCID iD for the corresponding author in Editorial Manager on papers submitted after December 6th, 2016. Please ensure that you have an ORCID iD and that it is validated in Editorial Manager. To do this, go to ‘Update my Information’ (in the upper left-hand corner of the main menu), and click on the Fetch/Validate link next to the ORCID field. This will take you to the ORCID site and allow you to create a new iD or authenticate a pre-existing iD in Editorial Manager. Please see the following video for instructions on linking an ORCID iD to your Editorial Manager account: https://www.youtube.com/watch?v=_xcclfuvtxQ

3) We note you have included a table to which you do not refer in the text of your manuscript. Please ensure that you refer to Tables 1 and 2 in your text; if accepted, production will need this reference to link the reader to the Tables.

4)  Thank you for stating the following financial disclosure:

 [The funders had no role in study design, data collection and analysis, decision to

publish, or preparation of the manuscript.].

Reviewers' comments:

5. Review Comments to the Author

Reviewer #1: The report presents the natural convection boundary layer flow of a viscous incompressible fluid with temperature-dependent viscosity and temperature-dependent thermal conductivity in the presence of exothermic catalytic chemical reaction along a curved surface. I recommend major revision due to the following facts.

1. In Eq. (3), the authors added (u^2/2x)n to the convective acceleration term.

Comment: What does the new term means? How did you derive it?

2. After Eq. (6), it was written that n = P(x) + Q(x). Whereas () is the wall temperature function and the () is the body shape function.

Comment: The authors refer Readers to check "Pop, I. and Takhar, H. S. Free convection from a curved surface. Journal of Applied Mathematics

and Mechanics, 73 (1993), pp. 534 -539" for details.

Comment: This is not acceptable. Update the research methodology with the derivation of such momentum equation.

3. Research question is needed to harmonize the contribution of the report to the body of knowledge.

Comment: Research questions are needed at the end of the introduction section. Research questions are needed. Note that the results in this report are typical answers to unknown questions. This is true because the manuscript provides some powerful answers to unknown questions. Note that the research questions must connect the title to the analysis of results, and conclusion. This would guide authors not to generate many results that are not consistent to provide insight. The author should update the manuscript with appropriate and relevant research questions at the end of the introduction section. This would guide the author to structure logical analysis of results. Logical questions are expected. This would help readers to link what is known in the literature with the novelty of this study.

4. The similarity variables presented as Eq. (7) are dimensional. The unit of the first, u, is m^0.5 while that of v is m^{-1/4}, and that of y is also dimensional. Hence, the variables used to non-dimenzionalize the governing equation are not appropriate.

Reviewer #2: Recommendation: Minor revision

Authors should revise the manuscript according to the following comments.

• Nomenclature is must.

• Some grammatical and typo mistakes are found in the manuscript.

• The governing differential equations describing the flow are non-linear. Is the solution obtained unique?

• Specify the applications of considered physical model or geometry.

• Introduction section should be made more concise to show previous work in the field. At present a lot of related researches are stated in the introduction. However, not sufficient analysis is presented. The authors should ask themselves: what are the problems with the presented researches? Why is the recent work needed? Hope these can improve the present work by following past articles.

• Effect of Joule heating on MHD non‐Newtonian fluid flow past an exponentially stretching curved surface

• Magnetohydrodynamic Cattaneo-Christov flow past a cone and a wedge with variable heat source/sink

• Heat and mass transfer in MHD Casson nanofluid flow past a stretching sheet with thermophoresis and Brownian motion

• Effect of asymmetrical heat rise/fall on the film flow of magnetohydrodynamic hybrid ferrofluid

• Simultaneous solutions for first order and second order slips on micropolar fluid flow across a convective surface in the presence of Lorentz force and variable heat source/sink

• Effect of thermal radiation on MHD Casson fluid flow over an exponentially stretching curved sheet

• Influence of non-uniform heat source/sink on the three-dimensional magnetohydrodynamic Carreau fluid flow past a stretching surface with modified Fourier’s law

• MHD Carreau Fluid Flow Past a Melting Surface with Cattaneo-Christov Heat Flux

• Physical aspects on unsteady MHD‐free convective stagnation point flow of micropolar fluid over a stretching surface

• Influence of viscous dissipation on MHD flow of micropolar fluid over a slendering stretching surface with modified heat flux model

• A non‐Fourier heat flux model for magnetohydrodynamic micropolar liquid flow across a coagulated sheet

Note: I want to see the revised version of the manuscript.

---

## [Author Response · Author response to Decision Letter 0]

13 Apr 2021

Response to Reviewers Comments 

(PONE-D-21-04681)

Effects of Temperature Dependent Viscosity and Thermal Conductivity on Natural Convection Flow along a Curved Surface in the Presence of Exothermic Catalytic Chemical Reaction

Reviewer #1: The report presents the natural convection boundary layer flow of a viscous incompressible fluid with temperature-dependent viscosity and temperature-dependent thermal conductivity in the presence of exothermic catalytic chemical reaction along a curved surface. I recommend major revision due to the following facts.

In Eq. (3), the authors added (u^2/2x)n to the convective acceleration term.

Comment: What does the new term means? How did you derive it.

Response: During dimensionalization of the momentum equation we have equation of the form 

u ∂u/∂x+v ∂u/∂y+u^2/2 1/g_x (dg_x)/dx=γ_μ (∂θ/∂y)(∂u/∂y) +(1+γ_μ θ)((∂^2 u)/(∂y^2 ))+θ+ϕ,

Here, as 1/x ® (P(x ®)+Q(x ®))=1/g_x (dg_x)/(dx ® )

and P(x ® )+Q(x ® )=n and thus 1/x ® (n)=1/g_x (dg_x)/(dx ® )

u ∂u/∂x+v ∂u/∂y+u^2/2x n=γ_μ (∂θ/∂y)(∂u/∂y) +(1+γ_μ θ)((∂^2 u)/(∂y^2 ))+θ+ϕ, 

2. After Eq. (6), it was written that n = P(x) + Q(x). Whereas () is the wall temperature function and the () is the body shape function.

Comment: The authors refer Readers to check "Pop, I. and Takhar, H. S. Free convection from a curved surface. Journal of Applied Mathematics and Mechanics, 73 (1993), pp. 534 -539" for details.

Comment: This is not acceptable. Update the research methodology with the derivation of such momentum equation.

Response: The detail derivation of n=P(x)+Q(x) is given in below steps:

Since we know that 

d/dx (lnx)=1/x

and 

d(lnx)=1/x dx

which implies that 1/(d(lnx))=x/dx. (1) 

Also consider d/dx (ln g_x )=1/g_x (dg_x)/dx

which implies d(ln g_x )=1/g_x dg_x. (2)

 Now combining the above two equations (1) and (2), we have

 (d lng_x)/(d lnx)=x/g_x (dg_x)/dx.

Now using the non-dimensionless variables, we may write the above equations as

(d lng_x)/(d lnx ®l)=(x ®l)/g_x (dg_x)/(dx ®l)

which implies that

(d lng_x)/(d lnx ®l)=x ®/g_x (dg_x)/(dx ® )

and thus we have 1/x ® (d lng_x)/(d lnx ®l)=1/g_x (dg_x)/(dx ® ) (3)

Also, we consider d/dx (lnT_w )=1/T_w (dT_w)/dx=0

which implies that d lnT_w=0

and thus, we can write (d lnT_w)/(d lnx ®l)=0 (4)

Now from equation (3) and (4), we have

 1/x ® ((d lnT_w)/(d lnx ® l)+(d lng_x)/(d lnx ® l))=1/g_x (dg_x)/(dx ® )

which can also be written as 1/x ® (P(x ®)+Q(x ®))=1/g_x (dg_x)/(dx ® ) (5)

Here, P(x ®), the wall temperature function and Q(x ®) the body shape function, are defined as

 P(x ® )=(d lnT_w)/(d lnx ® l), Q(x ® )=(d lng_x)/(d lnx ® l) . (6) 

3. Research question is needed to harmonize the contribution of the report to the body of knowledge.

Comment: Research questions are needed at the end of the introduction section. Research questions are needed. Note that the results in this report are typical answers to unknown questions. This is true because the manuscript provides some powerful answers to unknown questions. Note that the research questions must connect the title to the analysis of results, and conclusion. This would guide authors not to generate many results that are not consistent to provide insight. The author should update the manuscript with appropriate and relevant research questions at the end of the introduction section. This would guide the author to structure logical analysis of results. Logical questions are expected. This would help readers to link what is known in the literature with the novelty of this study.

Response: The overall objectives of this research are to develop the mathematical model to study and compare the different modes of coupling the exothermic and catalytic chemical reaction via momentum, energy and mass concentration equation. Model predictions are used to assess the effects of different parameters on conversion of exothermic catalytic chemical reaction at heated curve generated by tangential component of acceleration due to gravity. The proposed model is used as the guidelines for the selection of a suitable coupling to achieve the desired applications. A numerical technique Finite Difference Method in conjunction with primitive variable formulation is used to investigate the coupling of exothermic reaction with catalyst particles. Furthermore, parametric effects of heated wall and mass concentration along the curved surface are studied and highlighted graphically and as well as in tabular form. 

4. The similarity variables presented as Eq. (7) are dimensional. The unit of the first, u, is m^0.5 while that of v is m^{-1/4}, and that of y is also dimensional. Hence, the variables used to non-dimenzionalize the governing equation are not appropriate.

Response: In the section of numerical analysis, we have used primitive variable formulation given as below:

 u=x^(1⁄2) U(X,Y),v=x^((-1)⁄4) V(X,Y), x=X,y=x^(1⁄4) Y, 

 θ=Θ(X.Y), C=Φ(X,Y). 

 It is pertinent to mention that primitive variable are dimensionless, we use this transformation to get primitive form of partial differential equations, please see red terms in each equation 

 After using the aforementioned transformations, we have the following form of system of equations:

U/2+X ∂U/∂X-Y/4 ∂U/∂Y+∂V/∂Y=0,

[1/2+n/2] U^2+XU ∂U/∂X+(V-YU/4) ∂U/∂Y=γ_μ ∂Θ/∂Y ∂U/∂Y+(1+γ_μ Θ) (∂^2 U)/(∂Y^2 )+Θ+Φ,

XU ∂Θ/∂X+(V-YU/4) ∂Θ/∂Y=1/Pr [ξ(∂Θ/∂Y)^2+(1+ξΘ) (∂^2 Θ)/(∂Y^2 )]+βλ^2 X^(1⁄2) (1+nγΘ)exp((-E)/(1+γθ))Θ,

XU ∂Φ/∂X+(V-YU/4) ∂Φ/∂Y=1/Sc (∂^2 Φ)/(∂Y^2 )+λ^2 X^(1⁄2) (1+nγΘ)exp((-E)/(1+γθ))Φ.

Reviewer #2: Recommendation: Minor revision

Authors should revise the manuscript according to the following comments.

• Nomenclature is must.

Response: The nomenclature has been included in the revised manuscript.

• Some grammatical and typo mistakes are found in the manuscript.

Response: In the revised manuscript we have correct the grammatical issues.

• The governing differential equations describing the flow are non-linear. Is the solution obtained unique?

Response: Please if you focus on the graphs and boundary conditions

U_(i,j)=0, V_(i,j)=0,Θ_(i,j) =1, Φ_(i,j)=1 at Y_j=0

 U_(i,j)→0, 〖 θ〗_(i,j)→0, Φ_(i.j)→0 as Y_j→∞ 

In these graphs and all other graphs given in this study, the solutions are satisfying by the boundary conditions, it is evident of the correctness of the obtained results. 

• Specify the applications of considered physical model or geometry.

Response: The proposed model is used as the guidelines for the selection of a suitable coupling to achieve the desired applications. The curved shaped geometry is used to design many problems of civil engineering as pressure barrier.

• Introduction section should be made more concise to show previous work in the field. At present a lot of related researches are stated in the introduction. However, not sufficient analysis is presented. The authors should ask themselves: what are the problems with the presented researches? Why is the recent work needed? Hope these can improve the present work by following past articles.

• Effect of Joule heating on MHD non‐Newtonian fluid flow past an exponentially stretching curved surface

• Magneto hydrodynamic Cattaneo-Christov flow past a cone and a wedge with variable heat source/sink

• Heat and mass transfer in MHD Casson nanofluid flow past a stretching sheet with thermophoresis and Brownian motion.

• Effect of asymmetrical heat rise/fall on the film flow of magneto hydrodynamic hybrid Ferro fluid

• Simultaneous solutions for first order and second order slips on micro polar fluid flow across a convective surface in the presence of Lorentz force and variable heat source/sink.

• Effect of thermal radiation on MHD Casson fluid flow over an exponentially stretching curved sheet

• Influence of non-uniform heat source/sink on the three-dimensional magneto hydrodynamic Carreau fluid flow past a stretching surface with modified Fourier’s law

• MHD Carreau Fluid Flow Past a Melting Surface with Cattaneo-Christov Heat Flux

• Physical aspects on unsteady MHD‐free convective stagnation point flow of micro polar fluid over a stretching surface

• Influence of viscous dissipation on MHD flow of micro polar fluid over a slandering stretching surface with modified heat flux model

• A non‐Fourier heat flux model for magneto hydrodynamics micro polar liquid flow across a coagulated sheet

Note: I want to see the revised version of the manuscript.

Response: Most relevant papers are added to enrich introduction (see in introduction ref [23] - [30]) in the revised manuscript.

---

## [Decision Letter · Decision Letter 1]

28 Apr 2021

PONE-D-21-04681R1

Effects of Temperature Dependent Viscosity and Thermal Conductivity on Natural Convection Flow along a Curved Surface in the Presence of Exothermic Catalytic Chemical Reaction

PLOS ONE

Dear Dr. Ali,

Thank you for submitting your manuscript to PLOS ONE. After careful consideration, we feel that it has merit but does not fully meet PLOS ONE’s publication criteria as it currently stands. Therefore, we invite you to submit a revised version of the manuscript that addresses the points raised during the review process.

We look forward to receiving your revised manuscript.

Kind regards,

Naramgari Sandeep, Ph.D

Academic Editor

PLOS ONE

Journal Requirements:

Reviewers' comments:

Reviewer's Responses to Questions

6. Review Comments to the Author

Reviewer #1: The report presents the natural convection boundary layer flow of a viscous incompressible fluid with temperature-dependent viscosity and temperature-dependent thermal conductivity in the presence of exothermic catalytic chemical reaction along a curved surface. I recommend minor revision due to the following facts.

Q1. In Eq. (3), the authors added (u^2/2x)n to the convective acceleration term.

Comment: What does the new term means? How did you derive it.

In the report, the derivation presented by the Authors is in isolation. You are expected to derive it as a convective acceleration of momentum equation.

Q2. After Eq. (6), it was written that n = P(x) + Q(x). Whereas () is the wall temperature function and

the () is the body shape function.

This is the exact contribution of the report to the body of knowledge and it should be introduced under the introduction section. In fact, a paragraph is needed to announce this novelty.

Q3. How do you study a fluid flow along a vertical surface without buoyancy forces? This is true because there is nothing like the associated dimensionless parameter called buoyancy parameter or Grashof number.

Q4. Before Eq. 7, was defined as the tangential component of acceleration due to gravity. What is r in the definition?

Reviewer #2: The revisions are Good but some references are not arranged properly. Hence I recommend for possible publication.

---

## [Author Response · Author response to Decision Letter 1]

14 May 2021

Reply to Review PONE-D-21-04681R1

Effects of Temperature Dependent Viscosity and Thermal Conductivity on Natural Convection Flow along a Curved Surface in the Presence of Exothermic Catalytic Chemical Reaction PLOS ONE

Reviewer #1: The report presents the natural convection boundary layer flow of a viscous incompressible fluid with temperature-dependent viscosity and temperature-dependent thermal conductivity in the presence of exothermic catalytic chemical reaction along a curved surface. I recommend minor revision due to the following facts.

Q1. In Eq. (3), the authors added (u^2/2x)n to the convective acceleration term.

Comment: What does the new term means? How did you derive it. 

In the report, the derivation presented by the Authors is in isolation. You are expected to derive it as a convective acceleration of momentum equation.

Response: The detail derivation of momentum equation is given as below:

The dimensioned form of momentum equation in article is given as under

u ∂u/∂x+v ∂u/∂y=1/ρ ∂/∂y (μ ∂u/∂y)+g_x β_T (T-T_∞ )+g_x β_C (C-C_∞) (1)

Dimensionless variables 

x ®=x/l,y ®=y/l Gr^(1/4),u ®=u/U_s ,v ®=v/V_s Gr^(1/4),θ=(T-T_∞)/(T_w-T_∞ ) ,ϕ =(C-C_∞)/(C_w-C_∞ ) (2)

To convert the equation (1) into dimensionless form we use the dimensionless variables defined in equation (2), for this we find the term s one by one appeared in the equation (1)

u ∂u/∂x=U_s u ® (∂U_s u ®)/(∂x ®l)=(U_s^2)/l u ® (∂u ®)/(∂x ® )+〖U_s u ®〗^2/l (∂U_s)/(∂x ® )

u ∂u/∂x=(U_s^2)/l u ® (∂u ®)/(∂x ® )+〖U_s u ®〗^2/l ∂/(∂x ® ) (g_x β_T ΔT)^(1/2)

u ∂u/∂x=(U_s^2)/l u ® (∂u ®)/(∂x ® )+〖U_s u ®〗^2/l (g_x β_T ΔT)^(1/2) 1/(2√(g_x )) (dg_x)/(dx ® )

 u ∂u/∂x=(U_s^2)/l [u ® (∂u ®)/(∂x ® )+u ®^2/2 1/g_x (dg_x)/(dx ® )] (3)

Now

v ∂u/∂y=U_s v ®Gr^(-1/4) (∂U_s u ®)/(∂ly ®Gr^(-1/4) )=(U_s^2)/l v ® (∂u ®)/(∂y ® ) (4)

Now

1/ρ ∂/∂y (μ ∂u/∂y)=1/ρ [∂μ/∂y ∂u/∂y+μ (∂^2 u)/(∂y^2 )] (5)

Considering 

∂μ/∂y=∂/∂y μ_o (1+γ^* (T-T_∞))

∂μ/∂y=∂/(∂y ®lGr^(-1/4) ) (μ_o (1+γ^* ΔTθ))

∂μ/∂y=(μ_o γ^* ΔT)/(lGr^(-1/4) ) ∂θ/(∂y ® )

∂μ/∂y=(μ_o γ_T)/(lGr^(-1/4) ) ∂θ/(∂y ® ) (6)

∂u/∂y=(∂U_s u ®)/(∂y ®lGr^(-1/4) )=U_s/(lGr^(-1/4) ) (∂u ®)/(∂y ® ) (7)

(∂^2 u)/(∂y^2 )=(∂U_s u ®)/(∂y ®lGr^(-1/4) )=U_s/(lGr^(-1/2) ) (∂^2 u ®)/(∂y ®^2 ) (8)

Put eqs. (6)-(8) in eq. (5) we have 

1/ρ ∂/∂y (μ ∂u/∂y)=1/ρ [(μ_o γ_μ)/(lGr^(-1/4) ) ∂θ/(∂y ® ) U_s/(lGr^(-1/4) ) (∂u ®)/(∂y ® )+(μU_s)/(lGr^(-1/2) ) (∂^2 u ®)/(∂y ®^2 )]

1/ρ ∂/∂y (μ ∂u/∂y)=1/ρ [(μ_o γ_μ U_s)/(l^2 Gr^(-1/2) ) ∂θ/(∂y ® ) (∂u ®)/(∂y ® )+〖μ_o (1+γ_μ θ)U〗_s/(l^2 Gr^(-1/2) ) (∂^2 u ®)/(∂y ®^2 )]

1/ρ ∂/∂y (μ ∂u/∂y)=(μ_o U_s)/(ρl^2 Gr^(-1/2) ) [γ_μ ∂θ/(∂y ® ) (∂u ®)/(∂y ® )+(1+γ_μ θ)(∂^2 u ®)/(∂y ®^2 )]

Where γ_(μ=) γ^* ΔT

1/ρ ∂/∂y (μ ∂u/∂y)=(μ_o U_s)/(ρl^2 Gr^(-1/2) ) [γ_μ ∂θ/(∂y ® ) (∂u ®)/(∂y ® )+(1+γ_μ θ)(∂^2 u ®)/(∂y ®^2 )] (9)

Put the eqs. (3-4) and Eq. (9) in eq. (1)

(U_s^2)/l [u ® (∂u ®)/(∂x ® )+u ®^2/2 1/g_x (dg_x)/(dx ® )+v ® (∂u ®)/(∂y ® )]=(μ_o U_s Gr^(1/2))/(ρl^2 ) [γ_μ ∂θ/(∂y ® ) (∂u ®)/(∂y ® )+(1+γ_μ θ)(∂^2 u ®)/(∂y ®^2 )]+g_x β_T ΔTθ+g_x β_C ΔCϕ 

[u ® (∂u ®)/(∂x ® )+u ®^2/2 1/g_x (dg_x)/(dx ® )+v ® (∂u ®)/(∂y ® )]=(μ_o Gr^(1/2))/(ρlU_s ) [γ_μ ∂θ/(∂y ® ) (∂u ®)/(∂y ® )+(1+γ_μ θ)(∂^2 u ®)/(∂y ®^2 )]+(〖lg〗_x β_T ΔTθ)/(U_s^2 )+(〖lg〗_x β_C ΔTϕ)/(U_s^2 ) 

u ® (∂u ®)/(∂x ® )+u ®^2/2 1/g_x (dg_x)/(dx ® )+v ® (∂u ®)/(∂y ® )=(νGr^(1/2))/(lU_s ) [γ_μ ∂θ/(∂y ® ) (∂u ®)/(∂y ® )+(1+γ_μ θ)(∂^2 u ®)/(∂y ®^2 )]+(〖lg〗_x β_T ΔTθ)/(U_s^2 )+(〖lg〗_x β_C ΔCϕ)/(U_s^2 )

u ® (∂u ®)/(∂x ® )+u ®^2/2 1/g_x (dg_x)/(dx ® )+v ® (∂u ®)/(∂y ® )=(νGr^(1/2))/(lU_s ) [γ_μ ∂θ/(∂y ® ) (∂u ®)/(∂y ® )+(1+γ_μ θ)(∂^2 u ®)/(∂y ®^2 )]+(〖lg〗_x β_T ΔTθ)/(U_s^2 )+(〖lg〗_x β_C ΔCϕ)/(U_s^2 )

Where U_s=(g_x β_T ΔTl)^(1/2),Gr=(g_x β_T ΔTl^3)/ν^2 and U_sc=(g_x β_T ΔCl)^(1/2),Gr^*=(g_x β_T ΔCl^3)/ν^2 

So 

u ® (∂u ®)/(∂x ® )+u ®^2/2 1/g_x (dg_x)/(dx ® )+v ® (∂u ®)/(∂y ® )=(νGr^(1/2))/(lU_s ) [γ_μ ∂θ/(∂y ® ) (∂u ®)/(∂y ® )+(1+γ_μ θ)(∂^2 u ®)/(∂y ®^2 )]+(U_s^2 θ)/(U_s^2 )+(U_s^2 ϕ)/(U_s^2 )

u ® (∂u ®)/(∂x ® )+u ®^2/2 1/g_x (dg_x)/(dx ® )+v ® (∂u ®)/(∂y ® )=γ_μ ∂θ/(∂y ® ) (∂u ®)/(∂y ® )+(1+γ_μ θ)(∂^2 u ®)/(∂y ®^2 )+θ+ϕ (10)

Where the detail derivation of n=P(x)+Q(x) is given in below steps:

Since we know that

d/dx (lnx)=1/x

and

d(lnx)=1/x dx

which implies that1/(d(lnx))=x/dx. 

Also considerd/dx (ln g_x )=1/g_x (dg_x)/dx

which impliesd(ln g_x )=1/g_x dg_x. 

(d lng_x)/(d lnx)=x/g_x (dg_x)/dx.

Now using the non-dimensionless variables, we may write the above equations as

(d lng_x)/(d lnx ®l)=(x ®l)/g_x (dg_x)/(dx ®l)

which implies that

(d lng_x)/(d lnx ®l)=x ®/g_x (dg_x)/(dx ® )

and thus we have 1/x ® (d lng_x)/(d lnx ®l)=1/g_x (dg_x)/(dx ® ) 

Also, we consider d/dx (lnT_w )=1/T_w (dT_w)/dx=0

which implies that d lnT_w=0

and thus, we can write (d lnT_w)/(d lnx ®l)=0 

Now from equation (c) and (d), we have

1/x ® ((d lnT_w)/(d lnx ® l)+(d lng_x)/(d lnx ® l))=1/g_x (dg_x)/(dx ® )

which can also be written as 1/x ® (P(x ®)+Q(x ®))=1/g_x (dg_x)/(dx ® ) 

Here, P(x ®), the wall temperature function and Q(x ®) the body shape function, are defined as

P(x ® )=(d lnT_w)/(d lnx ® l), Q(x ® )=(d lng_x)/(d lnx ® l) . 

Now (10) becomes

u ® (∂u ®)/(∂x ® )+u ®^2/2(P(x ® )+Q(x ®))+v ® (∂u ®)/(∂y ® )=γ_μ ∂θ/(∂y ® ) (∂u ®)/(∂y ® )+(1+γ_μ θ)(∂^2 u ®)/(∂y ®^2 )+θ+ϕ (11) 

Now P(x ® )+Q(x ® )=n

Thus (11) becomes

u ® (∂u ®)/(∂x ® )+u ®^2/2 1/g_x (dg_x)/(dx ® )+v ® (∂u ®)/(∂y ® )=γ_μ ∂θ/(∂y ® ) (∂u ®)/(∂y ® )+(1+γ_μ θ)(∂^2 u ®)/(∂y ®^2 )+θ+ϕ 

Which is dimensionless form of momentum equation along curved surface, and n is the body shape parameter.

Q2. After Eq. (6), it was written that n = P(x) + Q(x). Whereas () is the wall temperature function and

the () is the body shape function. This is the exact contribution of the report to the body of knowledge and it should be introduced under the introduction section. In fact, a paragraph is needed to announce this novelty.

Response: In current paper we are investigated the natural convection flow over a two dimensional body of arbitrary geometric configuration in the presence of exothermic catalytic chemical reaction. The momentum, energy, and mass concentration equations are a general form suitable for laminar natural convection flows along curved surface in the inclusion of exothermic catalytic reaction. For body of arbitrary shape, the special case in which P(x ® )+Q(x ® )=n, thus the body shape parameter (index parameter) n has chosen 0<n≤1/2 which is main novelty of this work.

Note: This paragraph has inserted in the introduction section.

Q3. How do you study a fluid flow along a vertical surface without buoyancy forces? This is true because there is nothing like the associated dimensionless parameter called buoyancy parameter or Grashof number.

Response: Please if you see the dimensionless variables used to dimensionalize the flow model 

Dimensionless variables 

x ®=x/l,y ®=y/l Gr^(1/4),u ®=u/U_s ,v ®=v/V_s Gr^(1/4),θ=(T-T_∞)/(T_w-T_∞ ) ,ϕ =(C-C_∞)/(C_w-C_∞ ) 

 Here Grashof number is used as y ®=y/l Gr^(1/4), v ®=v/V_s Gr^(1/4) where Gr=(gβ∆Tx^3)/ν^2 , which is infact responsible for buoyancy force and it is used in dimensionalization (frequently available in literature (you can see Convective Heat Transfer by Ion Pop and D. B. Ingham for the cases of free or natural convection heat transfer).

 Dimensionless temperature variable θ=(T-T_∞)/(T_w-T_∞ ), it is the ratio of temperature difference. The problem under consideration is based on natural convection heat transfer, natural convection occurred because of the difference in density, and difference in density is due to the difference in temperature. The contribution of θ in momentum is obvious and sufficient answer to question.

Q4. Before Eq. 7, was defined as the tangential component of acceleration due to gravity. What is r in the definition?

Response: Here, x is the boundary layer coordinate along the curved surface while r and z are the horizontal and vertical coordinates of the points of the shape curve z=z(r) respectively. It is pertinent to mention that here flow is along the curved surface and dr/dz=0 means there is no flow inner side of the curve. The coordinate z is the vertical distance measure from the lower stagnation point (Please see fig. 1 flow configuration and coordinate system).

Reviewer #2: The revisions are Good but some references are not arranged properly. Hence I recommend for possible publication.

Response: We have focused on references, all the references are now in good order and well targeted to the problem under consideration.

---

## [Editor Report · Decision Letter 2]

17 May 2021

Effects of Temperature Dependent Viscosity and Thermal Conductivity on Natural Convection Flow along a Curved Surface in the Presence of Exothermic Catalytic Chemical Reaction

PONE-D-21-04681R2

Dear Dr. Ali,

We’re pleased to inform you that your manuscript has been judged scientifically suitable for publication and will be formally accepted for publication once it meets all outstanding technical requirements.

Kind regards,

Naramgari Sandeep, Ph.D

Academic Editor

PLOS ONE

Additional Editor Comments (optional):

The revised version is acceptable for publication.
---

## [Editor Report · Acceptance letter]

19 Jul 2021

PONE-D-21-04681R2 

Effects of Temperature Dependent Viscosity and Thermal Conductivity on Natural Convection Flow along a Curved Surface in the Presence of Exothermic Catalytic Chemical Reaction 

Dear Dr. Ali:

I'm pleased to inform you that your manuscript has been deemed suitable for publication in PLOS ONE. Congratulations! Your manuscript is now with our production department. 

Kind regards, 

on behalf of

Dr. Naramgari Sandeep 

Academic Editor

PLOS ONE